# Launching of hyperbolic phonon-polaritons in h-BN slabs by resonant metal plasmonic antennas

P. Pons-Valencia [1,11], F.J. Alfaro-Mozaz[2,11], M.M. Wiecha [2,8], V. Biolek [2,9], I. Dolado[2], S. Vélez [2,10], P. Li [2,3], P. Alonso-González [4], F. Casanova [2,5], L.E. Hueso [2,5], L. Martín-Moreno [1], R. Hillenbrand[5,6] & A.Y. Nikitin [5,7]

Launching and manipulation of polaritons in van der Waals materials offers novel opportunities for field-enhanced molecular spectroscopy and photodetection, among other applications. Particularly, the highly confined hyperbolic phonon polaritons (HPhPs) in h-BN slabs attract growing interest for their capability of guiding light at the nanoscale. An efficient coupling between free space photons and HPhPs is, however, hampered by their large momentum mismatch. Here, we show —by far-field infrared spectroscopy, infrared nanoimaging and numerical simulations— that resonant metallic antennas can efficiently launch HPhPs in thin h-BN slabs. Despite the strong hybridization of HPhPs in the h-BN slab and Fabry-Pérot plasmonic resonances in the metal antenna, the efficiency of launching propagating HPhPs in h-BN by resonant antennas exceeds significantly that of the non-resonant ones. Our results provide fundamental insights into the launching of HPhPs in thin polar slabs by resonant plasmonic antennas, which will be crucial for phonon-polariton based nanophotonic devices.

[1] Instituto de Ciencia de Materiales de Aragón and Departamento de Física de la Materia Condensada, CSIC-Universidad de Zaragoza, 50009 Zaragoza, Spain. [2] CIC nanoGUNE, 20018 Donostia-San Sebastián, Spain. [3] School of Optical and Electronic Information, Huazhong University of Science and Technology, 430074 Wuhan, China. [4] Departamento de Física, Universidad de Oviedo, 33007 Oviedo, Spain. [5] IKERBASQUE, Basque Foundation for Science, 48013 Bilbao, Spain. [6] CIC NanoGUNE and EHU/UPV, 20018 Donostia-San Sebastián, Spain. [7] Donostia International Physics Center (DIPC), 20018 Donostia-San Sebastián, Spain. [8] Present address: Physikalisches Institut, Johann Wolfgang Goethe-Universität, Max-von-Laue-Straße 1, 60438 Frankfurt am Main, Germany. [9] Present address: Institute of Physical Engineering, Brno University of Technology, Technická 2, Brno 616 69, Czech Republic. [10] Present address: Department of Materials, ETH Zürich, 8093 Zürich, Switzerland. [11] These authors contributed equally: P. Pons-Valencia, F. J. Alfaro-Mozaz. Correspondence and requests for materials should be addressed to L.M.-M. (email: lmm@unizar.es) or to R.H. (email: r.hillenbrand@nanogune.eu) or to A.Y.N. (email: alexey@dipc.org)

Low-dimensional van der Waals (vdW) materials have recently been attracting a substantial interest regarding photonic and optoelectronic applications since they support a variety of polaritons—oscillating dipolar excitations coupled to electromagnetic fields—that exhibit intriguing properties (such as nanoscale electromagnetic field confinement, tunability, or negative phase velocity)[1,2]. In hexagonal BN (h-BN), hyperbolic phonon polaritons (HPhPs) can propagate as ultra-confined rays within the mid-IR Reststrahlen bands (the spectral intervals between the transversal and longitudinal optical h-BN phonons, in which the transmission through the sample is strongly suppressed due to the negative real part of the dielectric permittivity), owing to the strongly anisotropic permittivity of h-BN. Due to their remarkably long lifetimes[3], HPhPs in h-BN (and especially, in isotopically enriched h-BN[4]) offer a strong potential for field-enhanced molecular vibrational sensing and strong coupling with molecular vibrations[5]. However, due to the large momenta of HPhPs, their direct excitation by an incident plane wave on an unstructured h-BN flake is not possible. To overcome the momentum mismatch and launch HPhPs, one can use the sharp tip of the near-field microscope[6], point and line defects[7], metallic edges[3,8], or edges of the h-BN flakes[8]. By structuring the h-BN flakes (e.g., cones[9], rods[10], or stripes[5]) one can also fabricate h-BN phononic antennas, where HPhPs exhibit Fabry-Pérot resonances, accessible with far-field illumination. Another promising strategy of launching HPhPs (being completely unexplored up to now) consists in using resonant metallic rod antennas. The use of the resonant metallic antennas is, in general, highly interesting for various opto-electronic applications, such as thermo-photovoltaics[11] or photocatalisis[11,12], due to their strong plasmonic response and large extinction cross-section[13]. Previously, resonant gold antennas (and gold resonator-based metamaterials) have been integrated with thin layers of isotropic polar dielectrics (SiO$_2$, SiC)[14–16], semiconductors (GaAs, GaN, ITO)[17–19], transparent conducting oxides (Al:ZnO, Ga:ZnO)[16], and monolayers of transition metal dichalcogenides (WS$_2$)[20], but not with a vdW material supporting long-lived HPhPs. Importantly, surface plasmon polaritons (SPPs) in the metallic antennas show strong coupling (hybridization) with the polaritons in thin slabs (except of graphene, where the interaction between graphene plasmon polaritons and SPPs in the resonant antenna is weak[21]). Due to the strong coupling, the near field of the resonant antenna can be partially screened, potentially hampering the launching of HPhPs. Therefore, to explore the launching of HPhPs in thin h-BN slabs by the resonant antenna and estimate the launching efficiency, it is crucial to get the understanding of the coupling between HPhPs and SPPs in the antenna.

Here, by means of Fourier-transform infrared spectroscopy (FTIR), scattering-type scanning near-field optical microscopy (s-SNOM) and numerical simulations we study the launching of HPhPs in thin h-BN slabs with resonant gold rod antennas. We find strong coupling between SPPs in the antenna and HPhPs in h-BN. At frequencies within the spectral gap emerging from the anti-crossing of SPPs and HPhPs dispersions, we manage to visualize the HPhPs modes emanating from the antenna and propagating along the h-BN slab. In spite of the suppression of the far-field antenna response in the h-BN Reststrahlen bands, we find that the HPhPs launching efficiency by the resonant gold rod antenna is significantly higher than that of non-resonant metallic launchers (e.g., non-resonant rods, small discs, or long stripes). Both numerical simulations and near-field experiments evidence that the launching efficiency has an optimum as a function of frequency, clearly related to the antenna resonance.

## Results

**FTIR spectroscopy of the individual Au antennas.** To study the spectral response of the gold rod antennas, we have performed FTIR transmission experiments (See Methods and schematics in Fig. 1a) of the individual Au antennas of different lengths placed on both h-BN flakes (55 nm thick) and on bare CaF$_2$ substrate. The schematics of the antenna on an h-BN slab is shown in Fig. 1b. Figure 1c (dashed curves) represents the difference transmission spectra, $\Delta T = T_0 - T$, normalized to its maximal value, max$(T_0 - T)$, for two antennas with the same geometries ($L = 2.29$ μm length, $w = 175$ nm width, and $h = 50$ nm height), one of which is placed on the CaF$_2$ substrate and the other one on the h-BN flake. Here, $T$ and $T_0$ are the infrared transmission for

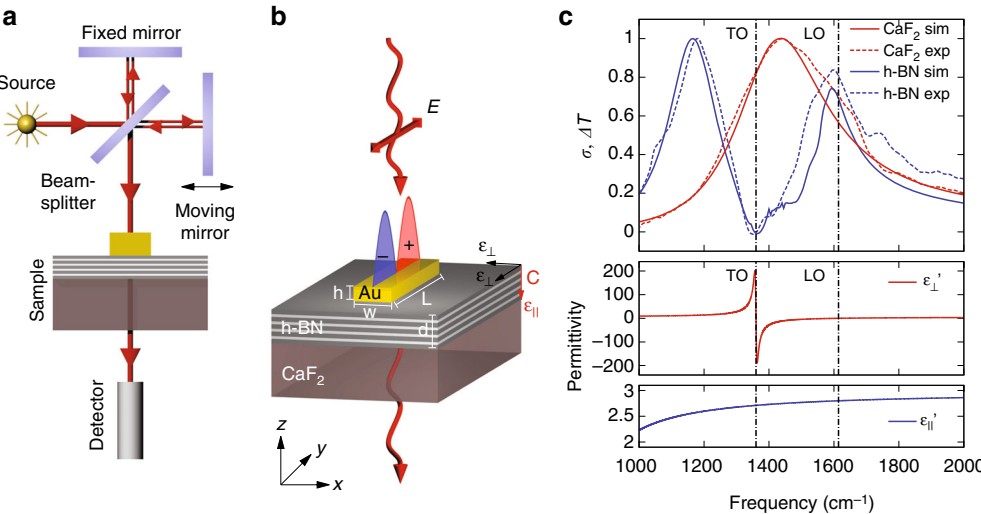

**Fig. 1** Infrared spectroscopy of gold rod antennas on top of an h-BN flake. **a** Schematics of the setup for the spectral far-field transmission measurements. **b** Schematics of the sample, illuminated by an incident plane wave: the gold rod antenna on top of the flake of h-BN placed on the CaF$_2$ substrate. The optical axis, C, and the dielectric permittivity tensor components, $\epsilon_\perp$ and $\epsilon_\parallel$, are indicated by the red and black arrows. **c** Top: normalized measured difference transmission spectra (dashed curves) and simulated extinction spectra (solid curves) for the antenna ($L = 2.29$ μm, $w = 175$ nm, and $h = 50$ nm) on CaF$_2$ (red curves) and on h-BN ($d = 55$ nm)/CaF$_2$ (blue curves). Bottom: real part of the perpendicular (red curve) and parallel (blue curve) components of the dielectric permittivity of h-BN as a function of frequency. Vertical dashed lines mark the positions of LO and TO phonons

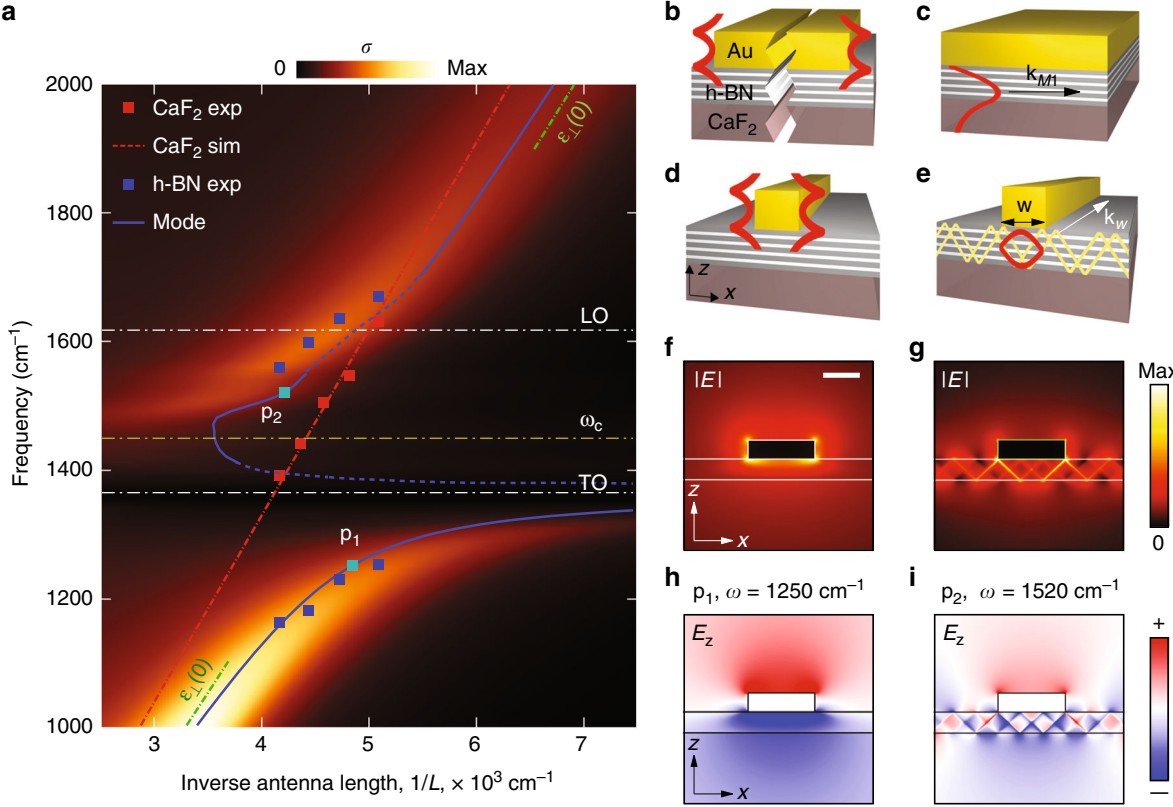

**Fig. 2** Infrared spectra of the gold rod antennas over h-BN slab for different antenna lengths and the mode analysis. **a** Colorplot: calculated extinction cross-section, $\sigma$, for the gold antenna on h-BN/CaF$_2$ as a function of $\omega$ and $1/L$. Symbols: maxima in the measured difference transmission spectra for the antennas on h-BN/CaF$_2$ (blue squares) and on CaF$_2$ (red squares). Blue solid and red dashed lines represent the dispersion of the mode for the infinite Au waveguide (with the cross-section of the antenna) on h-BN/CaF$_2$ and on CaF$_2$, respectively. The dashed parts of the blue curve (where as proper simulation of the dispersion was cumbersome) are a guide to the eye. Green dashed lines represent the asymptotes of the dispersion of the waveguiding mode. **b–e** Schematics of the infinite waveguides and the fields of the modes: wide Au stripe supporting edge SPP modes in **b**, h-BN slab below Au film supporting HPhP M1 mode (**c**), narrow Au rod supporting hybridized edge SPP-like mode and HPhP M1-like mode in **d**, **e**, respectively. The modes in **d**, **e** originate from the modes in **b**, **c**, respectively. **f–i** Simulated spatial field distributions of the modes of the narrow Au rod waveguide at the frequencies corresponding to the points p$_1$ (**f**, **h**) and p$_2$ (**g**, **i**) on the blue dispersion curve in **a**. The field distributions in **f**, **h** and **g**, **i** correspond to the schematics (**d**, **e**), respectively. Scale bar, 100 nm

the sample areas with and without the antenna, respectively (See Supplementary Note 1). The spectrum $\Delta T$ for the antenna on CaF$_2$ substrate (Fig. 1c, dashed red curve) clearly shows a resonance peak at 1440 cm$^{-1}$, corresponding to the excitation of the first-order dipolar SPP mode, as $L$ roughly matches half of the wavelength of light in CaF$_2$ substrate (see a more detailed analysis below). With increasing antenna length, the resonance peak position shifts linearly with $1/L$, as shown in Fig. 2a by the red squares (extracted from spectra of antennas of varying length). In stark contrast, the spectra of the antenna on the h-BN slab (Fig. 1c, dashed blue curve) manifests two peaks (at 1180 cm$^{-1}$ and 1600 cm$^{-1}$), with the position of the minimum between them nearly matching the frequency of the TO phonon (1360 cm$^{-1}$). The existence of two peaks at higher and lower frequencies than that of the TO phonon (with the separation between them larger than the width of each peak) points to strong coupling between the SPP mode of the antenna and HPhPs in h-BN. The signature of strong coupling is seen in the anti-crossing behavior of the resonance peak positions, extracted from the spectra of the antennas of different lengths on the h-BN slab (blue squares in Fig. 2a). Interestingly, the high-frequency peak crosses the LO phonon and penetrates the Reststrahlen band, while the low-frequency one remains below the TO phonon. Such behavior of the peaks is qualitatively different from the spectra of the antenna on an isotropic phononic material, where the surface phonon-polariton

branch appears[14]. In order to look into the origin of the peaks in the transmission spectra and clarify the role of the HPhP modes in h-BN slab, we perform their theoretical analysis.

**Theoretical analysis of the transmission resonances**. We compare the results of our transmission experiments with the extinction cross-section of the antennas found from full-wave electromagnetic simulations (see Methods). In Fig. 1c we present the extinction cross-section spectra, $\sigma$, normalized to their maximal value. We find that the extinction perfectly reproduces the experimentally measured transmission spectra for the antenna placed on both CaF$_2$ and h-BN (see solid red and blue curves in the upper panel of Fig. 1c). In the simulations we have not used any fitting parameters, but took the nominal values for the antenna geometry and h-BN slab thickness. Most importantly, in the simulated extinction as a function of both frequency, $\omega$, and antenna length, $L$, (colorplot in Fig. 2a), one can clearly recognize an anti-crossing feature, matching perfectly the experimental data (blue squares). To quantify the anti-crossing feature, we phenomenologically describe the coupling of the SPP antenna resonances and the HPhPs via a classical model of coupled harmonic oscillators (see Supplementary Note 4). From this model, we have extracted both the coupling strength, $g = 187 \pm 3$ cm$^{-1}$, and the losses associated with each oscillator ($\gamma_{SPP} = 555 \pm 28$ cm$^{-1}$ for the SPPs in the Au antenna

and $\gamma_{HPhP} = 28 \pm 7$ cm$^{-1}$ for the HPhPs in h-BN slab). According to the extracted values $g$, $\gamma_{SPP}$ and $\gamma_{HPhP}$, the two standard criteria for strong coupling, $g/(\gamma_{SPP} + \gamma_{HPhP}) > 0.25$ and $g/(\gamma_{SPP} - \gamma_{HPhP}) > 0.25$ are fulfilled[5], so that we can confidently conclude that the SPPs in the antenna are strongly coupled to HPhPs in the h-BN slab.

To understand the experimental and simulated transmission spectra and to clarify the origin of the strongly coupled modes, we performed a quasi-eigenmode analysis of an infinitely long metallic waveguide with a cross-section corresponding to that of our antenna, which is placed on an h-BN slab. Previously, a similar analysis allowed us to successfully interpret the Fabry-Pérot resonances in h-BN rod antennas[10]. In the quasi-eigenmode approach, the Au rod antenna can be seen as a truncated waveguide, in which propagating hybrid modes with wavevector $k_w(\omega)$ experience multiple reflections back and forth from the rod ends. As a result, longitudinal Fabry-Pérot resonances are built up, corresponding to constructive interference that occurs when

$$L \cdot k_w(\omega) + \varphi(\omega) = \pi n, \quad (1)$$

with $\varphi(\omega)$ being the frequency-dependent reflection phase of the vertical component of the electric field of the mode. The blue curve in Fig. 2a shows the Fabry-Pérot resonance condition for modes supported by the waveguide, for the fundamental longitudinal Fabry-Pérot resonance $(n = 1)$[22]. By analyzing the reflection of the mode from the end of the semi-infinite waveguide (see Supplementary Note 2), we found that the reflection phase, $\varphi(\omega)$, ranges between the values $\varphi_{min} = \pi/6$ and $\varphi_{max} = \pi/4$. The excellent agreement between the Fabry-Pérot resonance condition and the maxima in extinction (Fig. 2a) confirms that the resonances excited in the gold antenna can indeed be interpreted by Fabry-Pérot reflections of the hybrid electromagnetic mode of the truncated waveguide. Additionally, this excellent agreement allows us to conclude that the reflection phase of the mode in rod-like waveguides can be extracted directly from the extinction measurements of rod antennas.

Let us now analyze the waveguide modes in more detail in order to clarify their origin. Recall that the results presented in Fig. 2a are directly related to the dispersion relation of the modes, $k = k_w(\omega)$, via Eq. (1), so that in the following discussion we will refer to the dispersion relation, rather than to the Fabry-Pérot resonance condition. In order to understand the characteristics of the waveguide mode, we first consider an infinitely long gold rod on h-BN slab outside of the Reststrahlen band, where the contribution of the phonon oscillations is negligible. In this case, h-BN can be approximately considered as a non-dispersive material (in which the out-of-plane component of the dielectric permittivity, $\varepsilon_\parallel$, is not relevant because of the longitudinal character of the antenna resonance), characterized by dielectric permittivities $\varepsilon_\perp(\omega \to \infty)$ and $\varepsilon_\perp(\omega = 0)$ in the upper and lower frequency regions, respectively. In these frequency regions the dispersion of the mode of the infinitely long gold waveguide tends to its asymptotes (Fig. 2a, dashed green lines) corresponding to SPPs in the gold rod on h-BN/CaF$_2$. Because of the dielectric loading of the gold rod by the h-BN slab, the asymptotes of the dispersion stay separated from the dispersion of the SPP mode in the rod waveguide on CaF$_2$ substrate (Fig. 2a, red dashed line). The SPP dispersion perfectly matches the positions of the peaks in the measured extinction spectra of the gold antenna on CaF$_2$ substrate without h-BN slab (Fig. 2a, red squares). Due to the large values of the dielectric permittivity of Au, the dispersion of SPPs stays close to the light line in CaF$_2$, $k_w \simeq \sqrt{\varepsilon_{CaF2}}(\omega/c)$. The simulated field distribution at $\omega = 1250$ cm$^{-1}$ (point p1 on the blue curve in Fig. 2a) in Fig. 2f, h reveals the confinement of the field to the Au rod, thus confirming the plasmonic origin of the

waveguiding mode. In fact, this mode arises due to the hybridization of the edge SPP modes propagating along the two opposite edges of a wide Au rod[23,24] (schematics in Fig. 2b), similarly to HPhPs edge modes propagating along the edges of h-BN flakes[25]. When the width of the Au rod becomes comparable with the confinement of the SPP modes at the opposite edges of the rod, they hybridize into symmetric and anti-symmetric SPP modes, analogously to the HPhPs edge modes in h-BN waveguides (schematics in Fig. 2d)[10]. The spatial distribution of Re($E_z$) (Fig. 2h) reveals that the dispersion curve of our waveguide (Fig. 2a, blue curve) corresponds to the symmetric SPP edge mode.

Oppositely to the low and high-frequency ranges, inside the Reststrahlen band the waveguiding mode is dominated by propagating HPhPs in the h-BN slab. Indeed, the simulated field distribution at $\omega = 1520$ cm$^{-1}$ (point p$_2$ on the blue curve in Fig. 2a), reveals that the waveguiding mode is mainly concentrated in the h-BN slab below the metal rod (Fig. 2g, i). As typical for HPhPs in uniaxial crystal slabs, their fields have "zig-zag" patterns, manifesting rays that emerge from the corners of the Au rod[9,26,27] and reflect at the top and bottom of h-BN slab surfaces[28]. The zig-zag pattern can be represented as the superposition of waveguide modes M$n$, with $n = 0, 1, 2,...$[26,27], with both their wavelength and propagation length decreasing with $n$. Remarkably, the dispersion curve (Fig. 2a, blue curve) shows a back-bending (marked by the dashed yellow line in Fig. 2a), pointing to the existence of the cutoff frequency around $\omega_c = 1445$ cm$^{-1}$ in the lossless case. To explain the cutoff frequency, let us note that the lowest mode propagating in the h-BN slab below an infinite metal film (Au/h-BN/CaF$_2$, Fig. 2c), is M1, in contrast to the h-BN slab on a dielectric substrate, where the lowest mode is M0. It has the lowest wavevector, $k_{M1}$, and also the longest propagation length at a fixed frequency[6]. Therefore, we assume that the mode of the infinite rod-like waveguide should originate from the M1 mode below the infinite Au film. This assumption allows us to approximately represent the dispersion relation for the propagating waveguiding mode in an infinite rod-like waveguide (Fig. 2e) as $k_w^2 = k_{M1}^2 - (m\pi/w)^2$. In this formula, the second term presents the quantized momentum due to reflection of the mode M1 from the walls of the infinite metal rod (we neglect the reflection phase), due the impedance mismatch. For the lowest quantization order ($m = 1$), the condition $k_{M1} > \pi/w$, guarantees that $k_w$ is real and thus the mode can propagate along the waveguide. In contrast, when $k_{M1} < \pi/w$, then $k_w$ is imaginary and the mode cannot propagate. Therefore, the limiting condition $k_w = 0$ (or $k_{M1}(\omega_c) = \pi/w$) determines the cutoff frequency for the fundamental waveguiding mode. Its value $\omega_c = 1445$ cm$^{-1}$ perfectly matches with the position of the yellow dashed line in Fig. 2a, thus confirming our assumption that the waveguiding mode indeed originates from the mode M1.

Interestingly, we find that the field of the waveguiding mode in the Reststrahlen band extends well outside of the area of the Au rod (Fig. 2g, i), indicating a leakage of the waveguiding mode into HPhP in the h-BN slab. The leakage can be better seen in Fig. 3, where the spatial distribution of the simulated real part of the vertical electric field of the waveguiding mode, Re($E_z$), is shown both in the $x$–$z$ (Fig. 3a) and $x$–$y$ (Fig. 3b) planes. Generally, leakage takes place when a waveguide is placed on a medium supporting modes with wavevectors larger than that of the waveguiding mode, $k_w$[29,30]. For the h-BN slab on the CaF$_2$ substrate this condition is fulfilled for all the HPhPs M$n$ modes, $k_{Mn} > k_w$, so that the waveguiding mode leaks into all the modes M$n$ outside of the rod region. Since the losses of modes with higher $n$ increases, the M0 mode propagates the longest distance from the waveguide, as seen from the oscillations in Fig. 3, with their period matching the wavelength of the mode M0, $\lambda_{M0}$.

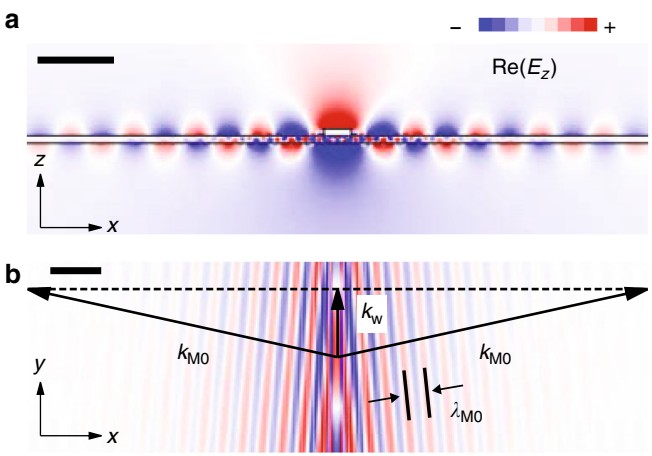

**Fig. 3** Leakage of the waveguiding HPhP mode below the Au rod into the HPhP M0 mode in the h-BN flake. **a, b** Spatial distribution of the z-component of the electric field in the z–x (**a**) and x–y (at the top surface of the h-BN slab) (**b**) planes, calculated at $\omega = 1520\ cm^{-1}$. The cross-section of the waveguide is that of the rod antennas. Scale bars, 0.5 μm in **a** and 1 μm in **b**

**s-SNOM visualization of the launched HPhPs**. We gain further insights into the properties of Au antennas on h-BN slabs by nanoimaging of the electric field created by the Au antennas employing s-SNOM. We illuminated the metallic tip of the s-SNOM with monochromatic infrared light of a frequency-tunable quantum cascade laser and recorded the amplitude of the tip-scattered field, $E_s$, with a pseudo-heterodyne Michelson inter-ferometer[31] as a function of the tip position (Fig. 4a). The signal is demodulated at higher harmonics of the tapping amplitude, $n\Omega$, yielding near-field images $s(x, y)$ (Fig. 4b) (for more details see Methods).

The near-field images of the Au antenna (Fig. 4c shows the topography image) on 55 nm thick h-BN slab at two frequencies are presented in Fig. 4e, h. An excellent agreement between the simulated field distribution, $|E_z(x, y)|$, (Fig. 4f, i) and measured near-field images, $|s(x, y)|$, (Fig. 4e, h) indicates that with s-SNOM we visualize the vertical electric field. In both simulated and experimental images one can clearly recognize fringes, similar to the ones created by linear gold edges[3] and gold disks[8] on h-BN slabs, or resonant gold rod antennas on graphene[21]. We attribute the fringes to the HPhPs launched by the gold antenna due to the generation of near fields with large momenta. These near fields, which are strongly confined to the antenna's edges are clearly seen in the simulated field distribution, $|E_z|$, of a Au rod antenna (illuminated by a plane wave with the electric field along the antenna axis) on a CaF₂ substrate (Fig. 4d). Note that, due to a much larger cross-section of the antenna compared to the tip apex, on one hand, and due to small reflection coefficient of the HPhPs from the antenna, on the other hand, the contribution of the tip-launched HPhPs to the near-field images is negligible (the tip-launched HPhPs appear as fringes with the distance between them equal to the HPhP half-wavelength). The HPhPs launched by the antenna into the h-BN slab are composed by different M$n$ modes, whose interference shows up as the "zig-zag" ray pattern (Fig. 4g, j). The mode M0 possesses the smallest wavevector and the longest propagation length, and its field dominates away from the antenna, while the zig-zag vanishes. The field of the M0 mode, $E_{z,M0}$, interferes with the incident field, $E_{z,i}$, resulting in fringes in the absolute value of the total field, $|E_{z,tot}|$, where $E_{z,tot} = E_{z,i} + E_{z,M0}$. The measured distance (along the green line in Fig. 4e) between the adjacent fringes (as shown in Fig. 5a by the blue horizontal arrow) as a function of frequency agrees with the wavelength of the

M0 HPhP mode, $\lambda_{M0}$. This agreement is seen from the comparison of $2\pi/\lambda_{M0}$ (Fig. 5c, open red dots) and the numerically calculated dispersion (Fig. 5c, red curve). Our near-field experiments and simulations thus demonstrate that the Au antenna indeed launches mainly the HPhP M0 mode into the h-BN slab, which interferes with the incident wave leading to the fringes in both the near-field images and spatial field distributions. We would like to note that the efficient launching of the M0 mode by the antenna can be also related with the leakage of the waveguiding mode, resonating in the antenna, into the M0 mode, as shown in Fig. 3. A detailed analysis of this speculation, however, goes beyond the scope of the current manuscript and should be conducted in future studies.

In order to extract the launching efficiency of the M0 mode from the experimental data, we estimate the relative amplitude of the M0 mode, $\eta$. To that end we analyze the visibility of the fringes in the near-field images, by calculating the difference between the maximal, A = max|s|, and minimal, B = min|s|, values of the scattered signal within the first fringe, divided by their sum, $\eta = (A − B)/(A + B)$ (Fig. 5a). The quantity $\eta^2$ presents the relative intensity of the M0 mode and therefore can be regarded as the launching efficiency. $\eta^2$ as a function of $\omega$ (Fig. 5d, open red dots) shows maximum at $\omega = 1430\ cm^{-1}$. To analyze the observed maximum, we simulate the part of the antenna absorption cross-section, contributed from the losses in the h-BN film, $\sigma_{M0}$ (launching cross-section, see Methods). Since the carrying-energy electromagnetic fields in the vicinity of the antenna are dominated by the M0 mode, the antenna-induced absorption in the h-BN slab scales as the energy flux of the M0 mode, $S_{M0}$, integrated over a closed surface (Fig. 5b). The positions of the maximal values of $\eta^2$ (Fig. 5d, open red dots) and $\sigma_{M0}$ (Fig. 5d, red curve) agree well, indicating that the launching efficiency extracted from the near-field images correlates with the energy absorbed around the antenna. Interestingly, the maxima of $\sigma_{M0}$ and total extinction cross-section, $\sigma$ (blue curve in Fig. 5d) occur at different frequencies. This difference in the frequency position can be explained by the spectral shift between the near- and far-field optical response, i.e., between $\sigma$ and local field enhancement, $|E|$, of resonant plasmonic antennas, leading to the shift in the peak intensities in the absorption and scattering, respectively[32] (See Supplementary Note 3). Furthermore, in Fig. 6, we present $\sigma_{M0}$ (normalized to the area of the antenna, $S_a = L \cdot w$) as a function of frequency and inverse antenna length (Fig. 6a). The colorplot clearly reveals the existence of an optimum in $\sigma_{M0}$ and proves that resonant antennas (red curve in Fig. 6b) launch HPhPs more efficiently than the non-resonant ones (both short and long ones, shown in Fig. 6b by the yellow and blue curves, respectively). In fact, for the resonant antenna, $\sigma_{M0}$ in its maximum is four times larger than $S_a$. Our additional analysis (Supplementary Fig. 4) also reveals that the $\sigma_{M0}/S_a$ of the resonant gold rod antennas is significantly larger than that of non-resonant launchers, particularly that of small gold disks and long gold stripes.

## Discussion

In conclusion, we employed infrared far-field spectroscopy and real-space nanoimaging to study individual metallic rod infrared optical antennas placed on h-BN slabs. We found the anti-crossing behavior of the antenna extinction spectra, revealing strong coupling between plasmon polaritons in the antenna and hyperbolic phonon-polaritons in h-BN, thus opening interesting possibilities for tailoring the optical response of polarizable phonon vibrations in low-dimensional materials. By a detailed quasi-mode analysis we showed that the observed extinction peaks can be explained by a hybridized waveguiding mode exhibiting longitudinal Fabry–Pérot resonances. This mode

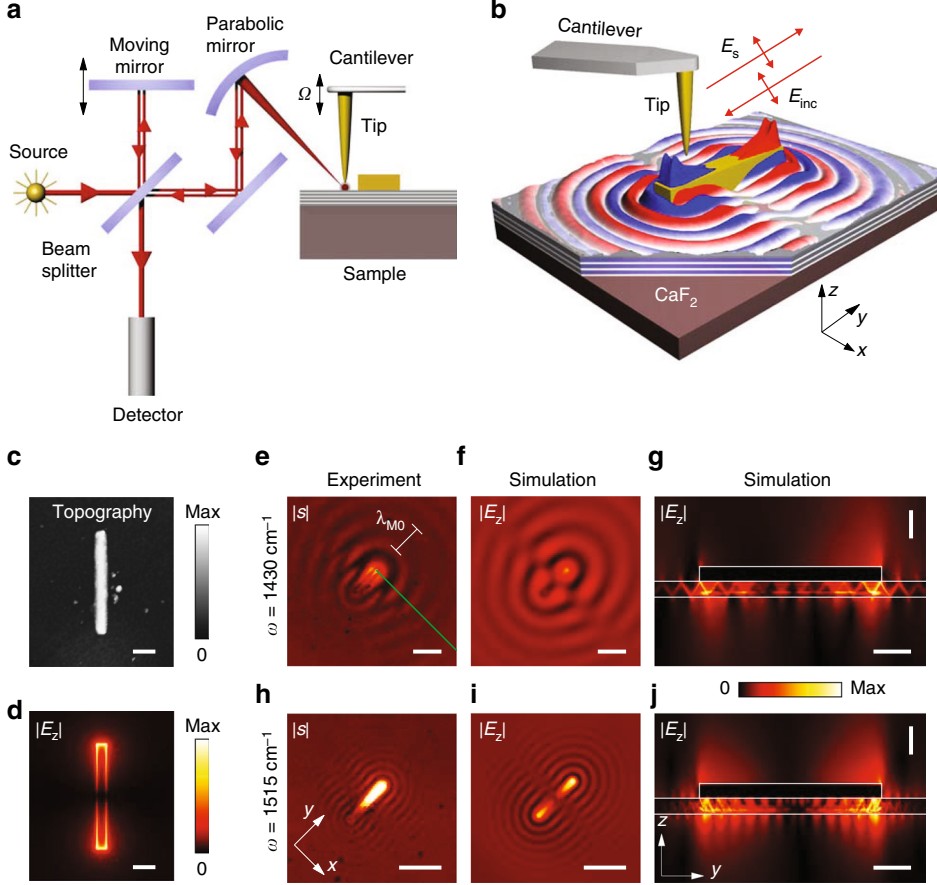

**Fig. 4** Near-field imaging of the HPhPs launched by the gold antenna. **a** Schematics of the s-SNOM setup. **b** Illustration of antenna launching of HPhPs. The spatial distribution of the near-field (shown by the red and blue colors) is adapted from the simulation of Re($E_z$). **c** Topography of the antenna. **d** Simulated near-field distribution, $|\mathbf{E}(x, y)|$, created by the rod antenna on CaF$_2$ (the field is taken at the top surface of the antenna). Scale bars in **c**, **d** are 0.5 μm. **e**, **h** Experimental near-field images. **f**, **i** Simulated near-field distribution $|E_z(x, y)|$ (taken 150 nm away from the h-BN slab). **g**, **j** Simulated near-field distribution $|E_z(z, y)|$ taken in the cross-section plane along the center of the rod antenna. In **e**–**g** $\omega = 1430$ cm$^{-1}$, while in **h**–**j** $\omega = 1515$ cm$^{-1}$. The scale bars in **e**–**i** are 2 μm and in **g**, **j** are 0.1 μm (vertical) and 0.5 μm (horizontal). The length of the antenna in all panels is $L = 2.29$ μm

has its field confined to the metallic antenna at frequencies outside of the Reststrahlen band (plasmon-polariton-like mode), while inside the Reststrahlen band its field is mainly concentrated inside the h-BN slab (phonon-polariton-like mode). Our study indicates that the waveguiding mode can leak into propagating phonon polaritons in the h-BN slab, thus enhancing the launching of the latter by the antenna. We visualized the antenna-launched hyperbolic polaritons with near-field imaging, and found a higher launching efficiency of the resonant antennas compared to that of the non-resonant launchers.

From a general perspective, our results pave the way to engineering metamaterials (or metasurfaces[33]) with unique optical properties composed of low-dimensional van der Waals crystals supporting hyperbolic polaritons and resonant metallic antennas. The latter can be interesting building blocks for the development of flat infrared molecular sensors or narrowband thermal emitters.

## Methods

**Electromagnetic simulations**. Full-wave numerical simulations were performed using finite-element method (COMSOL).

The extinction cross section, $\sigma = (A + S - A_0)/I_0$, was calculated as the sum of the Poynting vector flux through all the external boundaries of the simulation domain (scattering, $S$) and absorption in the domain volume ($A$) with the antenna, rested by the absorption in the domain volume without the antenna ($A_0$) and divided by the power flux density of the incident wave, $I_0$.

The launching cross-section, $\sigma_{M0}$, was calculated as $\sigma_{M0} = (A_a - A_0)/I_0$, where $A_a$ presents the absorption in the h-BN slab without the volume of the slab below the geometric area of the antenna.

Au dielectric permittivity was taken from ref. [34], CaF$_2$ is described by a polynomial $\varepsilon_{\mathrm{CaF2}} = 2.0683 - 0.018293\lambda_0 - 0.00015695\lambda_0^2 - 0.00016883\lambda_0^3$ (where the values of the free-space wavelength, $\lambda_0$, should be substituted in [μm]), while h-BN dielectric permittivity was modeled as described below.

*h-BN dielectric function*. h-BN dielectric permittivity tensor is modeled according to $\varepsilon_a^{\mathrm{hBN}} = \varepsilon_{a,\infty}\left(1 + \frac{\left(\omega_a^{LO}\right)^2 - \left(\omega_a^{TO}\right)^2}{\left(\omega_a^{TO}\right)^2 - \omega^2 - i\omega\gamma_a}\right)$ where $a = \parallel, \perp$ indicates the component parallel or perpendicular to the anisotropy axis. Used parameters are are $\varepsilon_{\parallel,\infty} = 2.95$, $\varepsilon_{\perp,\infty} = 4.90$, $\omega_\parallel^{TO} = 760$ cm$^{-1}$, $\omega_\parallel^{LO} = 825$ cm$^{-1}$, $\Gamma_\parallel = 2$ cm$^{-1}$, $\omega_\perp^{TO} = 1360$ cm$^{-1}$, $\omega_\perp^{LO} = 1614$ cm$^{-1}$, $\Gamma_\perp = 7$ cm$^{-1}$.

**Sample fabrication**. Sample preparation begins by isolating large and homogeneous h-BN flakes on a CaF$_2$ substrate. To that end, we first performed mechanical exfoliation of commercially available h-BN crystals (HQ graphene Co, N2A1) using blue Nitto tape (Nitto Denko Co., SPV 224P). Then, we performed a second exfoliation of the h-BN flakes from the tape onto a transparent poly-dimethylsiloxane stamp. After that, via both optical inspection and atomic force microscope characterization of the h-BN flakes on the stamp, high-quality flakes with large areas and required thickness were identified and transferred onto a CaF$_2$ substrate using the deterministic dry transfer technique[35].

We further fabricated Au antennas on both the CaF$_2$ substrate and on the h-BN flakes by means of high-resolution electron-beam lithography. We used Polymethyl methacrylate (PMMA) spin coated at 4000 rpm as the electron-sensitive polymer. The PMMA was subsequently covered by a 2-nm-thick layer of sputtered gold for facilitating the lithography on an insulating substrate. After the electron-beam exposure of the antenna structures, the gold was chemically etched (5 s immersion in KI/I2 solution) before developing the PMMA in methyl isobuthyl ketone:

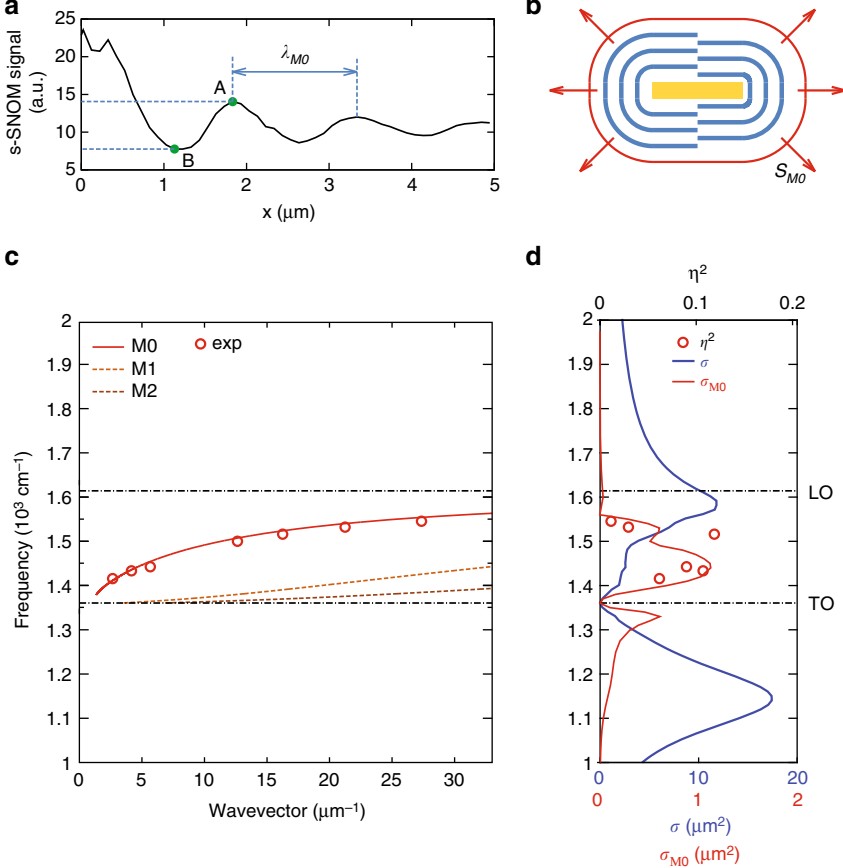

**Fig. 5** Launching efficiency of HPhPs by gold antennas. **a** Line scan along the green line in the near-field image of Fig. 4e. **b** Schematics of the launched HPhPs wavefronts (blue lines) and the Poynting vector integration contour (red line). **c** The dispersion of the first three HPhPs modes in the h-BN slab on top of $CaF_2$. The red dots represent the measured dispersion (extracted $2\pi/\lambda_{M0}$) from the line scans similar to **a**, taken at different frequencies. **d** The simulated launching cross-section for the resonant Au antenna, $\sigma_{M0}$ (red solid curve), and its extinction cross section, $\sigma$ (blue solid curve). The red dots render the launching efficiency for the resonant antenna extracted from the line scans similar to the one shown in **a**

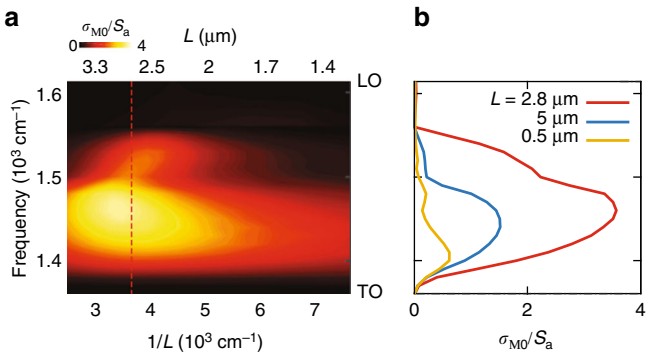

**Fig. 6** Launching efficiency of HPhPs for different gold antenna lengths. **a** Launching cross-section, $\sigma_{M0}$, normalized to the area of the antenna, $S_a$ as a function of both $1/L$ and frequency. The vertical dotted line mark the $1/L$ at which the red line shown in **b** has been extracted. **b** $\sigma_{M0}/S_a$ for the antennas with the lengths $L = 2.8\,\mu m$ (red line), $L = 5\,\mu m$ and $L = 0.5\,\mu m$. The vertical axes limits are the TO (bottom) and LO (top) phonon frequencies

isopropanol 1:3. Afterwards, 3 nm of a Ti stick layer was deposited by electron-beam evaporation onto the sample, followed by thermal evaporation of 40 nm of Au. Finally, the antennas were defined by lift-off in acetone overnight.

**FTIR measurements of single antenna**. Micro-spectroscopy transmission spectra were recorded with Bruker Hyperion 2000 IR microscope coupled to a Bruker Vertex 70 FTIR spectrometer (see schematics in Fig. 1a). We used a thermal source (globar) to generate normally incident unpolarized IR radiation. The spectral

resolution was $2\,cm^{-1}$. The aperture size was $10\,\mu m \times 10\,\mu m$ for an antenna on h-BN and $15\,\mu m \times 15\,\mu m$ for an antenna on $CaF_2$. In order to get rid of the noise in the measured spectra, they were smoothed by the moving average method using 25 elements.

**Near-field measurements with s-SNOM**. We used a commercial s-SNOM setup (Neaspec GmbH, Germany), in which the oscillating (at a frequency $\Omega \cong 270\,kHz$) metal-coated (Pt/Ir) AFM tip (ARROW-NCPt-50, Nanoworld) was illuminated by p-polarized mid-IR radiation. The radiation was generated by a quantum cascade laser (Daylight Solutions) and focused via a parabolic mirror onto both the tip and sample at an angle of 30 degrees with respect to the surface (p-polarization). The tip-scattered p-polarized light was recorded with a pseudo-heterodyne interferometer as a function of the tip position. The pixel size of the near-field images varies from 20 to 60 nm. To suppress the background scattering from the tip shaft and sample, the detector signal was demodulated (the oscillation amplitude was about 80 nm) at the frequencies of $2\Omega$ and $3\Omega$. The data shown in the manuscript are the average between demodulated signals at the second and third harmonics.

## Data availability
The data that support the findings of this study are available from the corresponding author on reasonable request.

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

## Acknowledgements

The authors acknowledge stimulating discussions with Prof. J. Aizpurua. The authors acknowledge financial support from the European Commission under the Graphene Flagship (GrapheneCore2), the Spanish Ministry of Science, Innovation and Universities (national projects MAT2017-88358-C3, MAT2015-65159-R, MAT2015-65525-R, RTI2018-094830-B-100, RTI2018-094861-B-100, and the project MDM-2016-0618 of the Marie de Maeztu Units of Excellence Program) the Marie Sklodowska-Curie individual fellowship (SGPCM-705960), the Basque Government (PhD fellowship PRE 2018 2 0253), and the project PIC201660E046 from CSIC. M.M.W. acknowledges support from the Konrad-Adenauer-Stiftung. P.A.G. acknowledges support from the European Research Council under Starting Grant 715496, 2DNANOPTICA.

## Author contributions

P.A.G., R.H., and A.Y.N. conceived the study. I.D. and S.V. fabricated the samples. F.C. and L.E.H. coordinated the fabrication. V.B. performed the far-field experiments coordinated by P.L. and M.M.W. performed the near-field experiments with the input of P.A.G. and R.H. P.P., and F.J.A. performed the full-wave simulations and analytical modelling. L.M.M., R.H., and A.Y.N. supervised the project. A.Y.N. wrote the manuscript with the input from all the co-authors. The latter also contributed to the scientific discussion and manuscript revisions.

## Additional information

**Competing interests:** R.H. is co-founder of Neaspec GmbH, a company producing scattering-type scanning near-field optical microscope systems, such as the one used in this study. The remaining authors declare no competing interests.

