## [Peer Review File · Nature Communications]

Reviewers' Comments:

Reviewer #1:

Remarks to the Author:

This is a very important and timely paper describing the launching of highly confined hyperbolic phonon polaritons (HPhPs) in hexagonal boron nitride (h-BN) slabs. Contrary to the very few other experimental studies of HPhPs in h-BN slabs (cited by the authors), the authors proposed a conceptually new and promising strategy of launching HPhPs. The proposed approach (being completely unexplored up to now) is based on the use of resonant metallic rod antennas.

The authors used far- and near-field spectroscopy/microscopy techniques to study individual metallic rod infrared optical antennas placed on h-BN slabs. The anti-crossing behavior of the antenna extinction spectra was found, demonstrating the strong coupling between the plasmon-polaritons in the antenna and hyperbolic phonon-polaritons in h-BN. These findings open exciting possibilities for tailoring the optical response of polarizable phonon vibrations in low-dimensional materials. By a detailed quasi-mode analysis, it has been showed that the observed extinction peaks can be explained by the hybridized waveguiding mode exhibiting longitudinal Fabry–Perot resonances. Thus, it is demonstrated that this mode is characterized by stronger field confinement to the metallic antenna outside of the Reststrahlen band, while inside the Reststrahlen band its field is mainly concentrated in the h-BN slab. This study indicates that the waveguiding mode can leak into propagating phonon-polaritons in the h-BN slab, thus enhancing the launching of the latter by the antenna.

The conclusions presented here are directly supported by near-field optical investigations that were performed to visualize the antenna-launched hyperbolic polaritons, interfering with the illuminating plane wave. A higher launching efficiency of the resonant antennas was found (compared to that of the non-resonant launchers).

The paper will be of wide interest for researchers in the nanosciences and should be published swiftly. I was particularly pleased to note the very careful description of the near-field optical results, which are adequately supported by theory. Therefore, I recommend this paper for publication.

Reviewer #2:

Remarks to the Author:

Infrared polaritonics are promising for IR sensors and directed transmission of IR near fields. A problem has been strong coupling external fields into the natural hyperbolic material. Crystal edges and wrinkles naturally lead to coupling of external fields to excite hyperbolic polaritons in hBN. Experimentally, the materials have been mainly studied by excitation using a near field scanning probe tip, which can provide the momentum to bridge the gap between the light line and the polariton dispersion curve. There have been a few additional examples where BN nanostructures have been made to support FP modes or similar that are generated by PhPs or other resonances of the BN. However, they have not lead to the kind of coupling that might be expected from a metallic antenna.

Therefore, the authors have explored the construction of a gold ribbon that has an infrared mode that is in or near the reststrahlen band of hBN. The generation of such a metal resonator is straightforward and the simulations and experimental data of the ribbon (or rod) response on CaF₂ are in agreement. The authors illustrate how the local fields of this gold nanostructure, when on hBN, should couple reasonably well with the crystal modes. Indeed, the coupling is strong, and two modes result from the strong coupling. The classical simulations result in plausible coupling strength and damping. These demonstrate that the goal of the paper has been achieved.

There is no doubt that this is an important paper - novel and extremely useful and creative; the coupling by the gold rod is strong and others in the field will want to explore what has been demonstrated.

I would encourage the authors to provide more background in explaining the stop band. In addition, a figure that reminds everyone of the relevant axes – epsilon parallel and perpendicular, in addition the anisotropy axis - will increase the understandability of the paper. An explanation of eqn 1, of why the phase of should be of vertical component of the electric field of the rod mode, would be more enlightening than highlighting the fact simply using italics. It would be useful to carry this aspect back to the permittivity axes of the material, and the coupling. Maybe a figure inset would be useful. These factors can be confusing.

Overall, I think this is an excellent piece of work that required at most minor revisions.

Response to Reviewer 1:

We thank the reviewer for the positive evaluation and for highlighting the novelty and the importance of our work.

Response to Reviewer 2:

We thank the reviewer for the positive report and for extremely valuable comments, which we address below in details.

I would encourage the authors to provide more background in explaining the stop band.

We provide the physical explanation of the “stop band” (Reststrahlen band), by modifying the text in Page 2 as follows:

hyperbolic phonon polaritons (HPhPs) can propagate as ultra-confined rays within the mid-IR Reststrahlen bands (the spectral intervals between the transversal and longitudinal optical h-BN phonons, in which the transmission through the sample is strongly suppressed due to the negative real part of the dielectric permittivity), owing to the strongly anisotropic permittivity of h-BN

In addition, a figure that reminds everyone of the relevant axes – epsilon parallel and perpendicular, in addition the anisotropy axis - will increase the understandability of the paper.

We have now added to Fig. 1b the schematics indicating the main crystal axis (C), as well as transversal and longitudinal components of the dielectric permittivity of h-BN. We have also added the following sentence to the caption of Fig. 1: .

(...) The optical axis, C, and the dielectric permittivity tensor components, ϵ_{\perp} and ϵ_{\parallel} , are indicated by the red and black arrows (...)

An explanation of eqn 1, of why the phase of should be of vertical component of the electric field of the rod mode, would be more enlightening than highlighting the fact simply using italics. It would be useful to carry this aspect back to the permittivity axes of the material, and the coupling. Maybe a figure inset would be useful. These factors can be confusing.

We apologize for the confusion introduced by the italic font. Actually, the reflection phase can be introduced for any component of the electromagnetic field. The choice of the field's component does not change the results. We considered the vertical component of the electric field simply for the consistence with other publications. We have now removed the italic highlight from the text.